# Experimental Study on Purification Effect of Biochemical Pool Model for Treatment of Pavement Runoff by Aquatic Plants

**Qinge Wang [1], He Cao [2,3], Huanan Yu [2,*], Luwei Zhao [1], Jinchan Fan [1] and Yingqing Wang [1]**

1   School of Civil Engineering, Central South University, Changsha 410075, China; wqecsu@csu.edu.cn (Q.W.);
    luweizhao@csu.edu.cn (L.Z.); 194811134@csu.edu.cn (J.F.); yingqingwang@csu.edu.cn (Y.W.)
2   National Engineering Laboratory for Highway Maintenance Technology, Hunan International Scientific and
    Technological Innovation Cooperation Base of Advanced Construction and Maintenance Technology of
    Highway, School of Traffic and Transportation Engineering, Changsha University of Science and Technology,
    Changsha 410114, China; caohe@smedi.com
3   Shanghai Municipal Engineering Design Institute (Group) CO.LTD Wuhan Branch, Wuhan 430000, China
*   Correspondence: huanan.yu@csust.edu.cn; Tel.: +86-137-1447-0593

**Abstract:** The road runoff after rainfall carries a lot of pollutants that could cause great harm to the water environment. A biochemical pool can be used as a treatment for the road runoff. In order to further improve the efficiency of road runoff treatment by biochemical pool and to evaluate the purification effect of the aquatic plants, two aquatic plants of *Iris pseudacorus* and *Myriophyllum verticillatum* were chosen in this research. The effect of different planting densities on the treatment of runoff pollutants and the planting mode by different aquatic plants were studied. The results show that both plants have the ability to remove the pollutants like chemical oxygen demand (COD), Zn, Cu, oil, and suspended solids (SS), and the ability is increased with the increase of planting density. The *Iris pseudacorus* is better than *Myriophyllum verticillatum* on the removal of Zn, while *Myriophyllum verticillatum* does better on the removal of Cu, oil, and SS. Combined planting mode can effectively improve the purification effect of COD and petroleum.

**Keywords:** road runoff; aquatic plant; pollution treatment; *Iris pseudacorus*; *Myriophyllum verticillatum*

## 1. Introduction

With the continuous development of transportation, the highway environmental problems brought by the operation of transportation infrastructure cannot be ignored [1,2]. For example, the highway surface runoff, which is formed after rainfall, will have a negative impact on the receiving water environment as a carrier of pollutants deposited on road surfaces.

Much research has been carried on this area around the world. Tony et al. [3] proposed that constructed wetlands can absorb suspended solids and pollutants in rainwater as an effective treatment measure for rainwater runoff pollution. Stotz [4] carried out extensive investigations and water analyses. Additionally, he found that when atmospheric precipitation (such as rainfall) takes place, pollution with numerous dissolved and undissolved substances occurs in surface water. Drapper et al. [5] evaluated the pollutants in runoff from road pavement surfaces following natural rainfall events in Brisbane, Australia. Moreover, runoff samples were detected to contain heavy metals, hydrocarbons, pesticides, and so forth. Besides, Holffman [6] in the, Narragansett Bay, USA, Gan et al. [7] in Guangzhou, China, Zhao [8] in Xi'an, China, also analyzed the pollutant components in road runoff, and the results indicated that the concentration of suspended solids (SS), chemical oxygen demand (COD), heavy

metals, petroleum, and other pollutants in road runoff from different countries and regions may exceed the grade II water quality standard for surface water.

When it comes to pollution, a lack of information on the load and characteristics of pollutants has led to insufficient reduction measures [9]. Byeon et al. [9] investigated the particle size and characteristics of rainfall-runoff pollutants collected from roads and combined sewers. Abrahams et al. [10] presented a case study on both water quality and productivity data from Brookside Farm, UK, using the constructed wetland ecosystem for sewage management and purification to absorb the high concentrations of nitrogen and phosphorous compounds in sewage. Gong et al. [11] explored the effects of bioretention on the removal of pollutants and explored the effects of runoff on plant growth and physiology. Muerdter et al. [12] verified that plants have different removal properties for suspended solids, nitrogen, phosphorus, and so forth. Additionally, rooting depth and planting density are important parameters, along with hydrologic impact, that require further study. Blanco [13] indicated totora's capacity to withstand high concentrations of a mixture of multiple pollutants and heavy metals. Tatiana A. et al. [14] conducted an experiment on the brown macro-algae harvested from the north coast of Portugal; several species of algae were used as natural cation exchangers for the treatment of zinc-containing rinse waters generated in the galvanizing process. The results showed the affinity of different algae to zinc ions. Abraham et al. [15] proposed an approach to recover nutrients and water from wastewaters before treatment by growing microalgae, suggesting the potential feasibility of using untreated, energetic-laden industrial wastewater along with carbon dioxide for microalgae biomass production. Law et al. [16] pointed out that green infrastructure (GI) is an economically, socially, and environmentally sustainable option for urban stormwater management. The system processes wastewater and places it in surface water. Nayak et al. [17] used domestic wastewater and flue gas as a culture medium and carbon source for improved microalgal biomass production, proving the purification ability of microalgal. Wu et al. [18] added cattail litter into a surface flow constructed wetland (SFCW) system to utilize as an additional carbon source for the denitrification process, which can improve nitrogen removal. These studies fall under the category of using plants for sewage treatment. Yu et al. [19,20] evaluated the adhesion and thermal properties of asphalt pavement, which could provide potential coating material for the application of pavement surface.

At present, engineering measures applied to the treatment of road surface runoff pollution mainly included vegetation control, detention pool, oxidation pond, constructed wetland, infiltration systems, and joint control measures [21]. Studies have shown that the joint control measures of aquatic plants combined with other measures can provide a good purification effect [22,23]. After 2000, the United States has put forward precise requirements on drainage standards and evaluation indicators within the scope of highway land, which protected the surface plants and their water purification functions, and has set up joint control devices for road runoff pollution [24].

Aquatic plants themselves have a good role in purifying water bodies, which have the characteristics of low investment, maintenance cost, simple operation, and low energy consumption. Since the 1960s, many scholars have successively adopted aquatic plants for sewage treatment. In 1953, Dr. Seidel [25] in Germany conducted the sewage purification experiment with artificial wetland and concluded that reed can effectively remove the inorganic and organic substances from sewage, and can also absorb and remove heavy metals from water. Tang [26] studied the application of a variety of aquatic plants in artificial wetlands and found that the experimental plants had a good removal effect on heavy metals. Tang et al. [27] studied nitrogen and phosphorus removal of 7 different aquatic plants, including *Acorus gramineus* in small constructed wetlands, and the results showed that different aquatic plants had different effects on nitrogen and phosphorus removal. Chen et al. [28] selected a variety of combinations of aquatic plants for experimental research in small constructed wetlands, and the results found that it had a good removal effect of heavy metals. Through the analysis and study on the purification of municipal sewage by duckweed, Steen et al. [29] found that it had a certain removal effect on biochemical oxygen demand (BOD), COD, and suspended solids. Wang et al. [30] found

different abilities of degradation of seven higher aquatic plants through the study of purifying water in simulated constructed wetlands.

The literature review found that effective purification effects of aquatic plants in road surface runoff have been verified by practice across the world. However, due to the difficulties in field facility experiments, there is still a lack of research on the characteristics of different influencing factors (such as planting density of aquatic plants, water pollution degree and water retention time) on the effect of water purification, which restrict the form of a scientific design method for the purification of plants polluted by runoff. In this paper, the role of aquatic plants in the treatment of pavement runoff is studied. A biochemical pool of aquatic plants, hereinafter referred to as a biochemical pool [31], is used as an example. The runoff decontamination and purification tests of the biochemical model pool was carried out under the conditions of different aquatic plants and different planting densities, and the removal and purification effect of COD, heavy metals, petroleum, and SS were obtained. The working principle of the biochemical pond is equivalent to the comprehensive action of the sewage disposal device and the constructed surface flow wetland. The difference between the biochemical pond and the constructed wetland is that it can be used as a sewage disposal device in case of water pollution caused by chemical leakage accident during road transportation, generally used as water environment protection device along the road in water source protection area.

## 2. Study Plan

### 2.1. Selection of Test Elements

The purification effect of the biochemical pond-scaled model for the treatment of road runoff by aquatic plants was studied. The main experimental elements involved were the types of aquatic plants, the collection (or preparation) of runoff from the road, and the setting of the biochemical pond.

### 2.1.1. Aquatic Plant Selection

Aquatic plants transport oxygen from the air to the roots, providing a living aerobic microenvironment for aerobic microorganisms around the roots of plants, enabling aerobic microorganisms to better decompose organic matters such as oil in water. At the same time, regions far away from the root area form oxygen-consuming regions, and BOD and COD in sewage can be reduced through oxygen-consuming metabolism [32]. Aquatic plants can absorb and accumulate heavy metals in sewage by themselves and can oxidize heavy metals in sewage by increasing the oxygen content in the root zone through oxygen pumping, thus reducing the pollutant content in water [33]. Aquatic plants block and absorb suspended particles of heavy metals through luxuriant branches and leaves and developed roots, and achieve the effect of purifying water quality through physical action [34].

In this paper, the plants selected were *Iris pseudacorus* and *Myriophyllum verticillatum*. *Iris pseudacorus*, also known as yellow calamus, belongs to the family Iridaceae. It is a multi-year aquatic plant [35] widely distributed in temperate and subtropical regions of the world. It grows all over China, has strong adaptability to the environment, strong tolerance to pollutants, and strong cold resistance. *Myriophyllum verticillatum* belongs to the perennial dicotyledonous submerged plant of haloragaceae, distributed in north and south China [36]. It grows in spring and reaches its maximum biomass in summer, can remove nitrogen, phosphorus, and other nutritive elements from water quickly; it is a strong water purification plant that can also absorb heavy metal elements.

### 2.1.2. Preparation of Experimental Runoff Water

It is difficult to control water quality by collecting runoff water from road surfaces and to meet the needs of a large number of tests with a long span. Road-deposited sediments (RDS) often contain elevated concentrations of inorganic and organic pollutants such as heavy metals, metalloids, and polycyclic aromatic hydrocarbons [37]. Road sediments (RDS) from a highway were collected and analyzed and found to be highly contaminated with organic matter, nutrients, and metals [38,39].

For this reason, the method of collecting the pavement sediment and preparing the water solution was adopted to carry out the experimental study [36].

The sediment formed on the road surface is the main source of road runoff pollution. Affected by wind and rain, about 90% of it is distributed within the range of 0–50 cm from curb [38,40–42]. All the samples collected in this paper are regarded as the pavement sediment within the range of 0–50 cm of the curb, which can be used as the typical representative of the main source of pavement runoff pollution. The large particles and sundry matter in the samples were removed, and an appropriate amount of them was weighed and added into a certain amount of tap water to be configured into different concentrations of water, so as to replace the actual pavement runoff water for research. The runoff wastewater was prepared by adding 300 g of collected road sediment into 160 L of tap water.

### 2.1.3. Biochemical Model Pond Test Chamber

The biochemical pool was based on the biochemical pool in a runoff treatment system [41] of a high-speed road in Guangzhou, China, as the prototype (as shown in Figure 1). The size of the prototype was 5 m in length, 10 m in width, and 2.5 m in depth. The biochemical pool has many functions such as vegetation control, retention pool, oxidation pond, and constructed wetland.

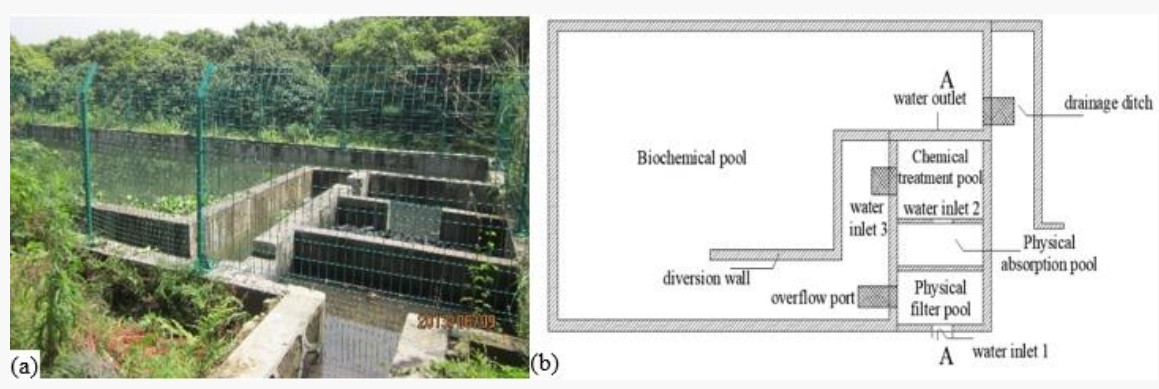

**Figure 1.** Biochemical pool prototype: (**a**) Biochemical pool on site; (**b**) Layout plan of biochemical pool

The theoretical analysis with an experimental test was used to obtain the effective purification mechanism of aquatic plants on road runoff. Since it is difficult to carry out multiple full-scale tests, the scaled model pond was used in this study [43].

According to the growth space of aquatic plants and the minimum water level requirement, the height ratio of the scaled model pond needed to be different from the horizontal geometric ratio. In the scaled model pond, the depth scale was named as $\lambda_h$ and horizontal length scale was named as $\lambda_l$, the value of $\lambda_h$ is less than $\lambda_l$, and $\lambda_h/\lambda_l$ is recorded as $\eta$. Fisher found that the variation ratio generally lies between 2 and 6, and many scholars also confirmed that the variation ratio of the abnormal model pond should be controlled within 5 in order for the results obtained by the model pond to be better applied in practice [43].

The horizontal scale of the biochemical pool was taken for 20:1, the depth scale was taken for 4:1, so $\eta$ was 5 in this case, which was between the scale of 2 and 6. The final height of $h$ was 62.5 cm, and width of $b$ was 50 cm, and length $l$ was 1.25 m of the scaled model pond. The model pond was made of PVC board, and the bottom was paved with about 10 cm thick of silt as the planting layer, and 1 cm of fine sand was added to prevent the water from flowing and driving the silt to flood. The model pond also had the inlet and outlet. In order to prolong the retention time of runoff in the pond and achieve the best biological purification effect, a certain length of folded diversion wall was designed at the inlet. During the test, water flowed in through the inlet and out through the outlet as shown in Figure 2 below.

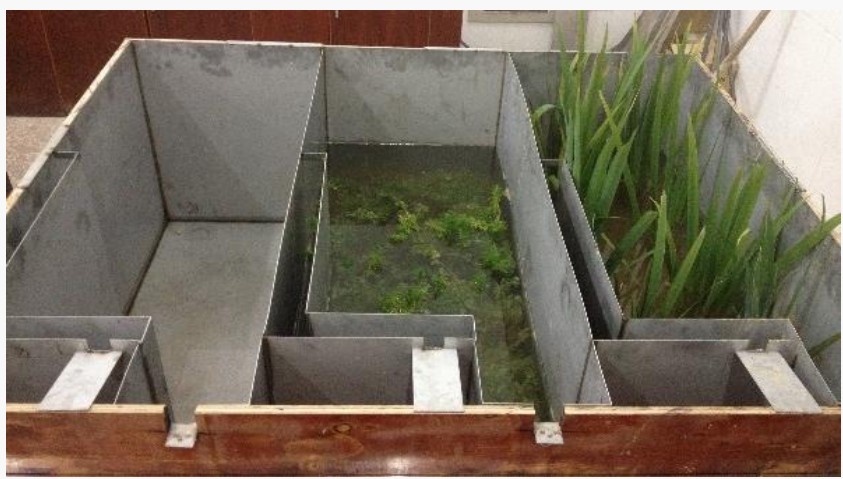

**Figure 2.** Biochemical model pond box.

## 2.2. Research Design and Method

In the same area, the concentration of pollutants in runoff from the rainy road is the highest at the beginning of the rainy season. The mechanism of the plants' purification effect is mainly through two parts; the first one is physical adsorption, the second one prevents pollutant diffusion. Advanced sewage processing is the following degradation of the trapped pollutants in the biochemical pool [39]. Therefore, to study the purification effect of aquatic plants in treating runoff pollution from the road surface, it is necessary to evaluate the purification effect of pollutant adsorption and diffusion during the period of runoff formation but we also need to analyze the purification ability of retentive degradation after rainfall. Based on the above situation, the whole experiment process was designed as shown in Figure 3 below.

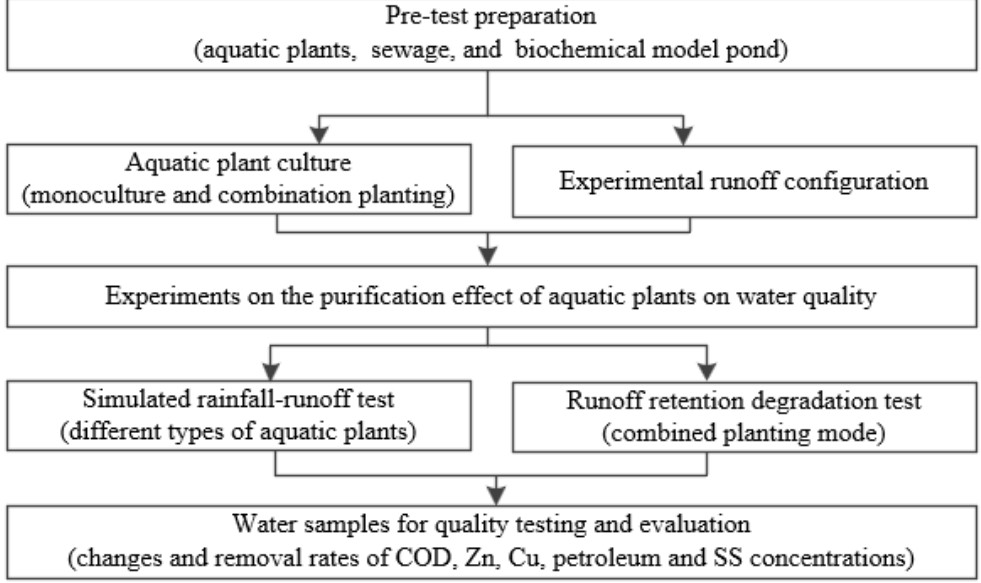

**Figure 3.** Flow chart of the experiment plan.

### 2.2.1. Cultivation of Aquatic Plants in Biochemical Model Pond

The aquatic plants were planted with certain planting density in the biochemical model tank according to the two modes: single plant cultivation and combined plant cultivation. The details are shown in Table 1. The water level was kept in the box at 5 cm from the overflow, and follow-up experiments were conducted after cultivation for one month. The tests included three different plans.

**Table 1.** Experimental parameter design.

| Experiments | Aquatic Plant Cultivation Mode | Planting Density |
|---|---|---|
| Simulated rainfall-runoff test | *Iris pseudacorus* monoculture model | 34 bushes/m$^2$, 18 bushes/m$^2$, and 9 bushes/m$^2$, 0 bushes/m$^2$ (blank group) |
| | *Myriophyllum verticillatum* monoculture model | 10 bushes/m$^2$, 5 bushes/m$^2$, and 2 bushes/m$^2$, 0 bushes/m$^2$ (blank group) |
| | Combination planting model (*Iris pseudacorus + Myriophyllum verticillatum*) | 34 clumps/m$^2$ + 10clumps/m$^2$ 18 clumps/m$^2$ + 5 clumps/m$^2$ 9 clumps/m$^2$ + 2 clumps/m$^2$ |
| Runoff retention degradation test | Combination planting model (*Iris pseudacorus + Myriophyllum verticillatum*) | 18cluster/m$^2$ + 5cluster/m$^2$ |

*Iris pseudacorus* **monoculture model.** The *Iris pseudacorus* used in the experiment is about 80 cm high and 2–3 buds/clumps. Generally, the recommended planting density is 20 to 25 clusters/m$^2$. According to the plane size of the biochemical pool, we planted *Iris pseudacorus* uniformly with the density of 34 bushes/m$^2$, 18 bushes/m$^2$, and 9 bushes/m$^2$ in the scaled pool. Meanwhile, a blank group was set without planting plants for a control test.

*Myriophyllum verticillatum* **monoculture model.** The *Myriophyllum verticillatum* used in the test is about 20 cm high and 25–30 buds/clumps. Generally, the recommended planting density is 4 to 6 clusters/m$^2$. *Myriophyllum verticillatum* was planted uniformly with a density of 10 bushes/m$^2$, 5 bushes/m$^2$, and 2 bushes/m$^2$ in the scaled pool. Meanwhile, a blank group without planting plants was set for a control test.

**Combination planting model.** *Iris pseudacorus* and *Myriophyllum verticillatum* were combined and planted according to three different densities (*Iris pseudacorus + Myriophyllum verticillatum*): 34 + 10 clumps/m$^2$, 18 + 5 clumps/m$^2$, and 9 + 2 clumps/m$^2$.

2.2.2. Test of Runoff Pollution Adsorption and Diffusion Purification Effects During Rainfall

After the rainfall has formed, the rainwater scours the road surface and converges on the side of the road to form road runoff. Road runoff is continuously injected into the treatment system. When the water level in the treatment system reaches the height of the overflow threshold, the overflow begins. Besides, the overflow water is directly discharged into the receiving water. The road runoff water flows into the biochemical pool, and through the physical adsorption of plants and preventing the diffusion of sewage pollutants in the pond, to achieve adsorption and diffusion purification of runoff pollution [34].

In the storm runoff, the concentration of pollutants reached a peak within 30 min after the formation of the road runoff, and the concentration is lower after that. Therefore, during the experiment, the entire continuous water injection duration was about 60 min, of which the overflow time was 30 min.

The experimental runoff sewage was taken to as the 0# water sample before water injection. At the rate of 800 mL/min, the experimental runoff sewage was injected, and time was started when the water overflowed from the water outlet. The overflow water samples (i.e., 1#, 2#, and 3# water samples) were collected for examination at 10, 20, and 30min, respectively. Among them, the 0# water sample test value was the initial concentration value, and the 1#, 2# and 3# water sample test values were used as the post-treatment concentrations at different moments during the rainwater runoff.

2.2.3. Experiment on Purification Effect of Remaining Runoff Polluted Water Degradation

At the end of the rainfall, after the inflow and outflow of the biochemical pool were completed, the pollutants trapped in the retention pool water were degraded by the aquatic plants in the biochemical pool, thereby purifying the water in the pool.

After the biochemical model pool for planting aquatic plants was filled with the configured runoff sewage, parallel water samples were collected and tested at 1, 8, 24, 48, and 72 h, respectively.

## 2.3. Evaluation Criteria

The evaluation criteria used for the water quality evaluation, including COD, petroleum, SS, and various heavy metals (Zn, Cu). The tested items, detection method, and main testing instruments were shown in Table 2 below. According to the China national sample storage standard (GB12997-91), the samples were sealed and stored at low temperatures with chemical reagents added. In addition, the examination should be completed within the specified time to ensure the accuracy and reliability of the analysis results.

**Table 2.** Water quality indicators and methods.

| No. | Items | Test Specification | Test Equipment |
|-----|-------|--------------------|----------------|
| 1 | COD | GB11914-89 | GDYS-201M Multi parameter water quality analyzer |
| 2 | Zn | GB7475-87 | GDYS-201M Multi parameter water quality analyzer |
| 3 | Cu | GB7475-87 | GDYS-201M Multi parameter water quality analyzer |
| 4 | Petroleum | SL93.2-94 | MAI-50G Oil content analyzer |
| 5 | SS | GB11901-89 | Oven, Electronic balance |
| 6 | pH value | GB5750-85 | GDYS-10SP Acidity tester |

The test data were processed and the calculation formula for the treatment rate of road runoff sewage was shown in Equation (1):

$$\text{Removal rate} = \frac{C_0 - C_1}{c_0} \times 100\% \tag{1}$$

where $C_0$ is the concentration of pollutants before treatment of road runoff sewage which means the initial concentration value in each group of experiments(mg/L), $C_1$ is the concentration of pollutants after road runoff sewage treatment which means the concentration after treatment at different moments (mg/L).

## 3. Results and Discussions

### 3.1. Effects of Planting Patterns of Aquatic Plants on Runoff Water Quality Purification

Different aquatic plants and different planting densities have different purification effects on water quality. The single planting and combined planting modes of aquatic plants were used to simulate the effects of runoff pollution adsorption and diffusion purification during rainfall. The experiment was conducted according to the above process, and the results were shown in Figures 4–6.

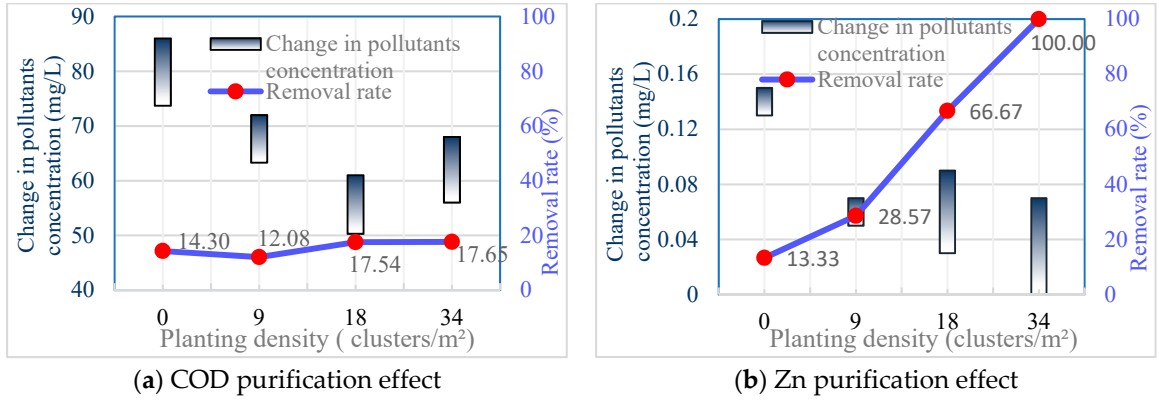

(**a**) COD purification effect　　(**b**) Zn purification effect

**Figure 4.** *Cont.*

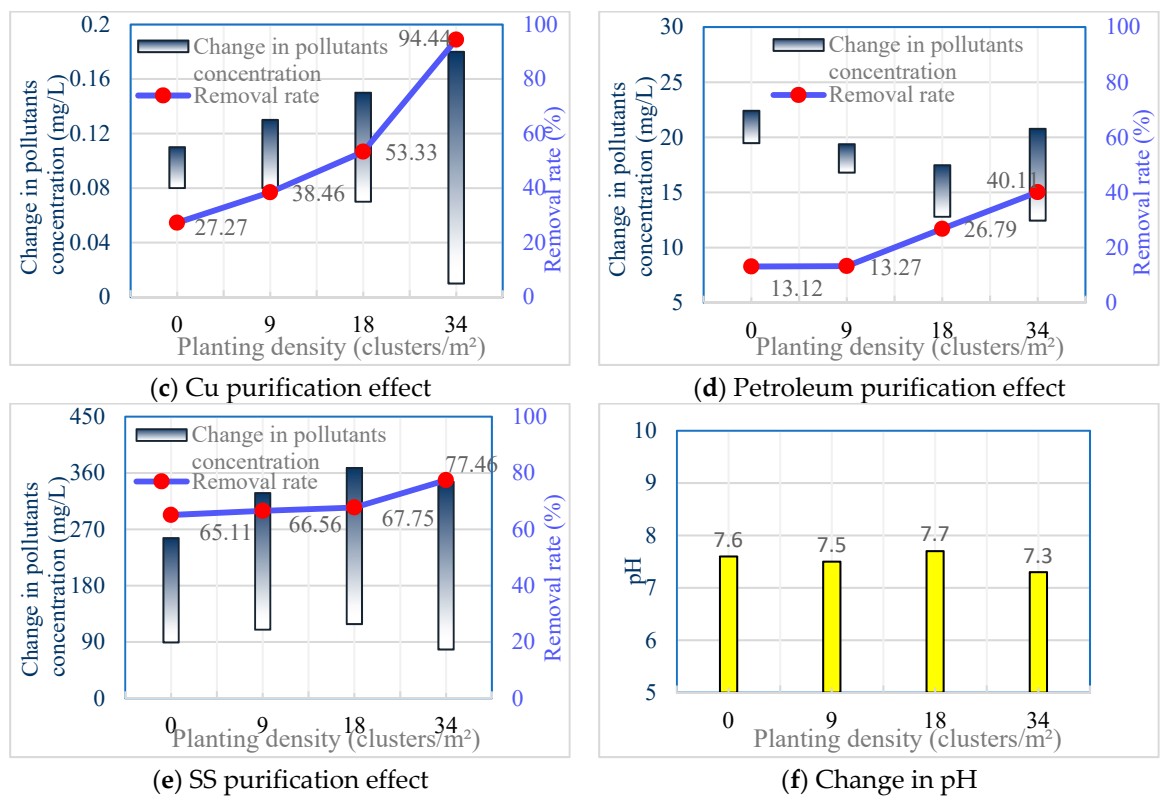

**Figure 4.** Variation of pollutant concentration in runoff water quality in *Iris pseudacorus* monoculture model.

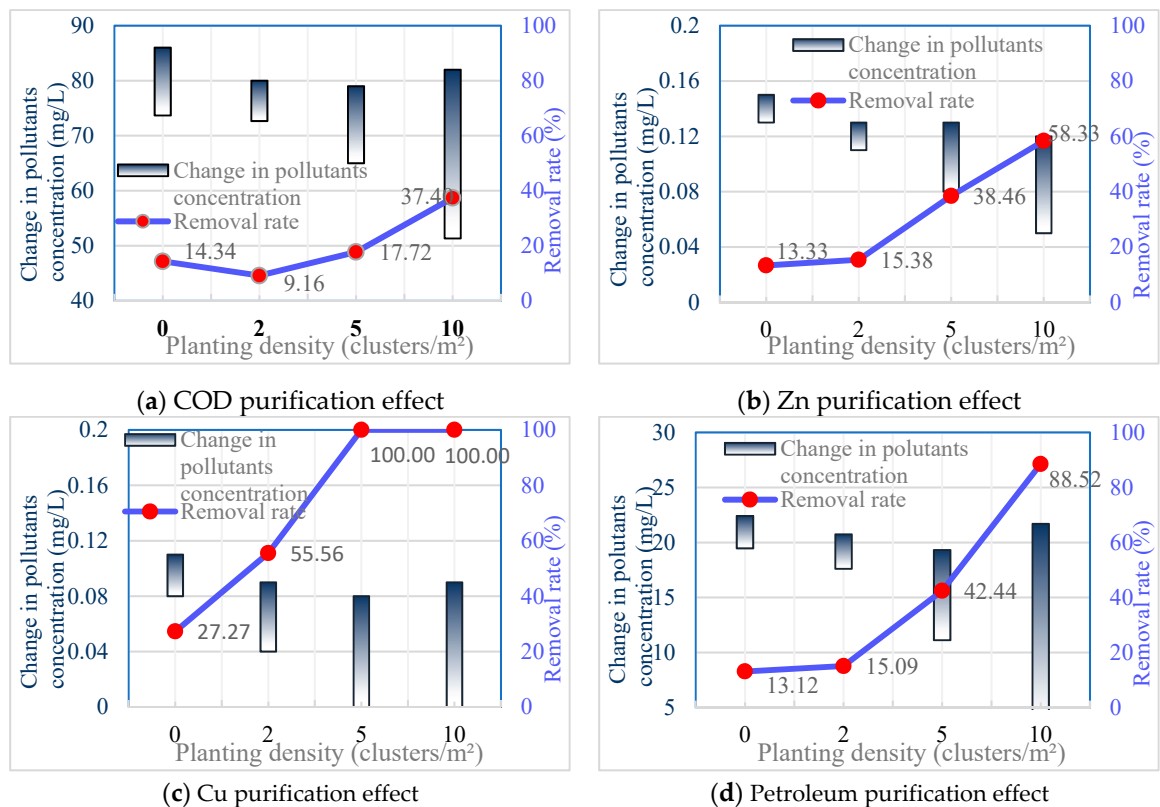

**Figure 5.** *Cont.*

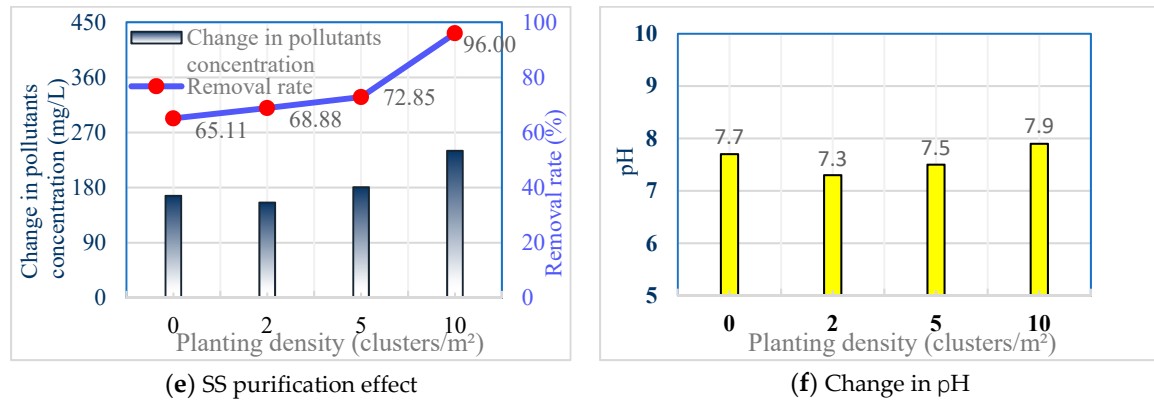

**Figure 5.** Variations of concentrations of pollutants when single-planted *Myriophyllum verticillatum.*

Under the given working conditions, the experiments were carried out on the purification of runoff water pollution by single planting of *Iris pseudacorus* and *Myriophyllum verticillatum*. The results in Figures 4 and 5 showed that both *Iris pseudacorus* and *Myriophyllum verticillatum* had different effects on the purification of pollutants in water. When the planting density of *Iris pseudacorus* varied from 9 to 34 clumps/m$^2$, the removal rates of COD, Zn, Cu, petroleum, and SS components of suspended solids ranged from 12.08–17.65%, 28.57–100.00%, 38.46–94.44%, 13.27–40.11%, and 66.56–77.46%, respectively. When the planting density of *Myriophyllum verticillatum* varied from 2 to 10 clumps/m$^2$, the removal rates of COD, Zn, Cu, petroleum, and SS components of suspended solids ranged from 9.16–37.40%, 15.38–58.33%, 55.56–100.00%, 15.09–88.52%, 68.88–96.00%, and the treatment effect of *Iris pseudacorus* and *Myriophyllum verticillatum* on pollutant components in water quality increased with the increase of planting density, which indicated that its purification effect had a density effect.

In the treatment effect of single plant cultivation, the removal rate of COD by both *Iris pseudacorus* and *Myriophyllum verticillatum* was not high, generally within 20%; the two plants also have different absorption of heavy metals due to their different growth needs, and the treatment effect of *Iris pseudacorus* on Zn was better than that of *Myriophyllum verticillatum*, which can better remove Cu in sewage. *Myriophyllum verticillatum* was more effective in trapping petroleum and suspended solids than that of *Iris pseudacorus*, and the removal effect of petroleum and suspended solids was more obvious, which was related to the lush branches and leaves in the water and the netlike morphology of stems and leaves formed under the water, so that the surface area of leaves in contact with water was much higher than that of *Iris pseudacorus*.

In the blank group experiment (0 clumps/m$^2$), the removal rates of COD, Zn, Cu, petroleum and SS components of suspended solids were 14.34%, 13.33%, 27.27%, 13.12%, and 65.11%, respectively, which indicated that the pollutant components in the experiment had certain purification effect caused by sedimentation. pH value has been maintained between 7–8 during the experiment, and aquatic plants had little effect on it. As the experiment process was less than 30 min, within a short period of time, plants had negligible effect on the pH regulation of water quality.

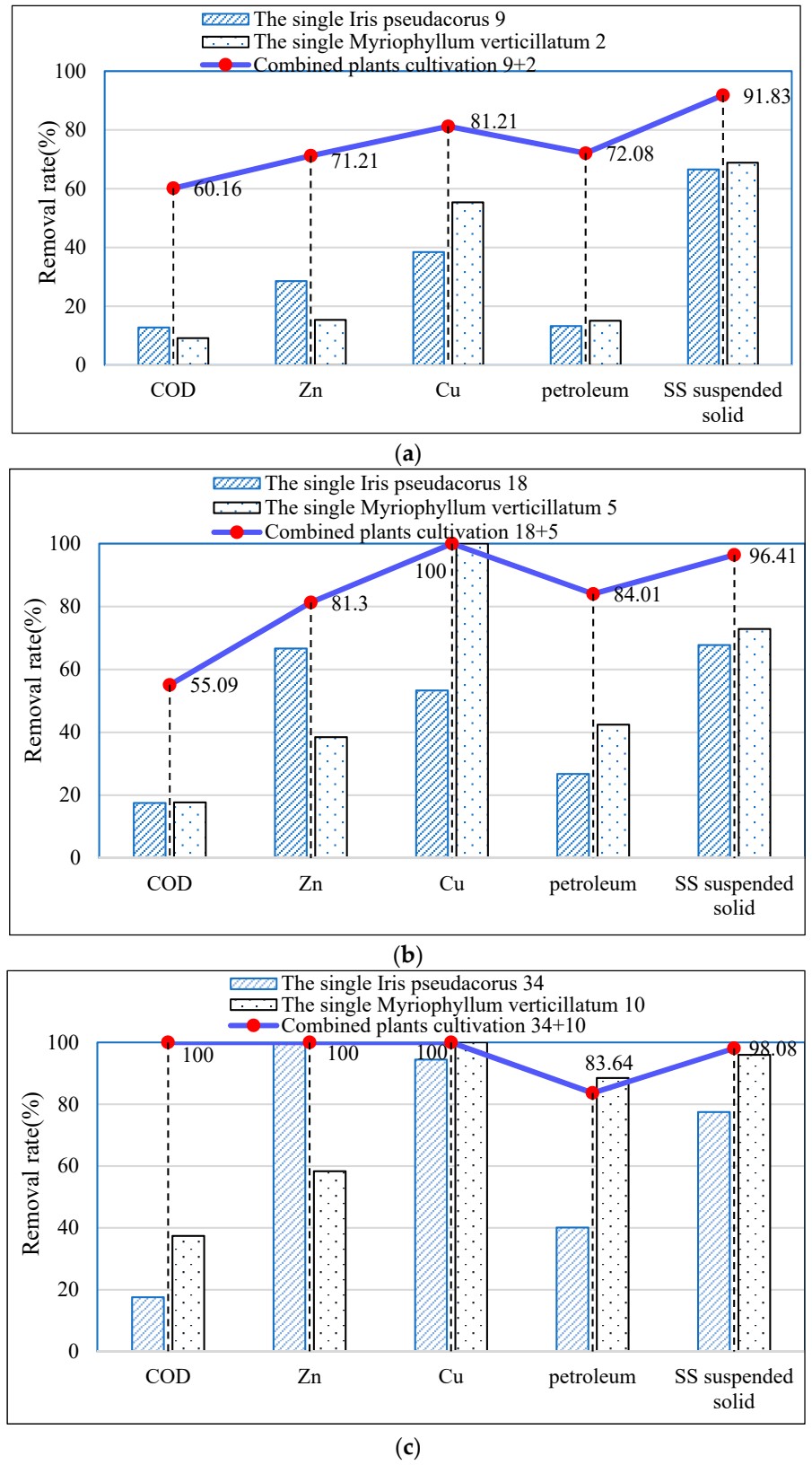

**Figure 6.** Comparison for purification effect of combined planting on various pollutants: (**a**) Purification effect of 9 bushes/m$^2$ of *Iris pseudacorus* + 2 bushes/m$^2$ of *Myriophyllum verticillatum*; (**b**) purification effect of 18 bushes/m$^2$ of *Iris pseudacorus* + 5 bushes/m$^2$ of *Myriophyllum verticillatum*; (**c**) purification effect of 34 bushes/m$^2$ of *Iris pseudacorus* + 10 bushes/m$^2$ of *Myriophyllum verticillatum*.

According to the results on the purification of runoff water pollution components under the two-plant combination model of *Iris pseudacorus* and *Myriophyllum verticillatum* in Figure 6, it can be known that the treatment of aquatic plants under the combined planting mode of the three plants had an obvious effect on several pollutants; under the three different combinations of densities, the removal rate of COD ranged from 55.09% to 100.00%, the removal rate of Zn ranged from 71.21% to 100.00%, the removal rate of Cu ranged from 81.21% to 100.00%, and the removal rate of petroleum ranged from 72.08% to 84.01%, the removal rate of SS ranged from 91.83% to 98.08%. Compared with the single planting mode, the plant combination can exert the advantages of different plants in the purification of pollutants, and the purification rate was higher than that of the single planting plant; the purification effect of COD and petroleum was especially significant. Take the *Iris pseudacorus + Myriophyllum verticillatum* (18 + 5 clumps/m$^2$) mode for example; the combined mode purification rate of COD and petroleum removal was higher than the purification effects of two types of single planting. At the same time, in the combined experiment with different densities, it can be seen from the changes in the concentration of various pollution in the combined planting experiment also showed a density effect—the higher the planting density, the better the purification effect.

### 3.2. Dynamic Influence on Runoff Pollution Degradation and Purification Process

According to the commonly recommended planting density of *Iris pseudacorus* and *Myriophyllum verticillatum*, *Iris pseudacorus* was taken as 18 cluster/m$^2$ and *Myriophyllum verticillatum* was taken as 5 cluster/m$^2$ to cultivate in a biochemical model pond. And the biochemical model pond was filled with runoff sewage. The dynamic effects of pollution components on the degradation and purification time under the action of aquatic plants in the biochemical pond. The experimental results are shown in Figure 7 and Table 3.

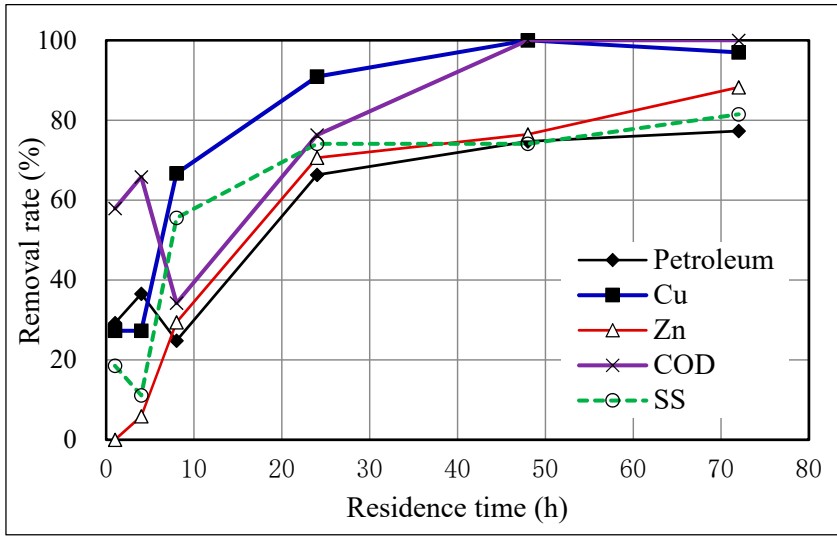

**Figure 7.** The purification effect of plants on pollutants changes with time.

**Table 3.** The change of pH value with time during the experiment.

| Residence Time | 1 h | 4 h | 8 h | 24 h | 48 h | 72 h |
|---|---|---|---|---|---|---|
| pH value | 7.9 | 8.0 | 8.0 | 8.1 | 7.7 | 7.4 |

The above experiment results showed that the pollutant removal effect of runoff water within 4 h of detention pond fluctuated greatly, indicating that after the sewage was injected into the pond, the diffusion of pollutants in the pond was still relatively disordered, the obtained water samples had relatively large randomness of pollutant content; the removal rate of each pollutant showed a steady increase after standing for 4 h; the removal rate of pollutants almost reached more than 80% after 72 h

of standing, which indicated a better performance with the extension of the residence time. In the experiment, the removal rate of each pollution component had an increasing trend with the extension of the residence time within 72 h, and the purification efficiency was the highest at 8–24 h, after which the pollution removal and purification efficiency decrease.

During the experiment, after the pH value was allowed to stand for 72 h, the carbon dioxide produced by the respiration of the plants reacted with the water to form a weakly acidic substance, which lowered the pH value of the water [44], indicating that the combined plant had a certain regulating effect on the pH of the water.

## 4. Conclusions

Different planting modes were adopted to carry out the biochemical tank scale model purification experiment of treating road runoff by *Iris pseudacorus* and *Myriophyllum verticillatum*, and the following conclusions were obtained.

Under different planting modes, the purification effect of *Iris pseudacorus* and *Myriophyllum verticillatum* on the pollutants of COD, Zn, Cu, petroleum, and SS suspended solids all had density effect, the purification effect generally increased with the increase of planting density.

Plant types and planting patterns had different purification effects on different pollutant components. In the treatment effect of the single plant planting pattern, the removal rate of COD by *Iris pseudacorus* or *Myriophyllum verticillatum* was not high. The treatment effect of *Iris pseudacorus* on Zn was better than *Myriophyllum verticillatum*, and *Myriophyllum verticillatum* can better remove Cu in sewage. Combined planting mode can effectively improve the purification effect of COD and petroleum.

In the experiment, the removal rate of various pollutants in runoff water by aquatic plants increased steadily after standing for 4 h. Additionally, a good removal effect with the extension of the retention time was shown, the purification rate was the highest at 8–24 h, after which the purification rate gradually decreases.

Since only two kinds of aquatic plants were selected as experimental objects in this paper, other plants or multiple plant combinations can be considered in future research to study the treatment effect of road runoff pollutants.

**Author Contributions:** Conceptualization, Q.W. and H.Y.; methodology, Q.W. and H.Y.; software, H.C.; validation, L.Z., J.F. and Y.W.; formal analysis, H.Y. and H.C.; investigation, H.Y.; resources, Q.W.; data curation, H.Y. and H.C.; writing—original draft preparation, H.C. and Y.W.; writing—review and editing, L.Z. and J.F.; visualization, H.C.; supervision, Q.W. and H.Y.; project administration, Q.W.; funding acquisition, Q.W. All authors have read and agreed to the published version of the manuscript.

**Funding:** The study was funded by major project of Guangdong provincial department of transportation: study on pollution law and prevention countermeasures of expressway surface runoff in the water-sensitive areas in south China (zc-gcky-2-02).

**Conflicts of Interest:** The authors declare no conflict of interest.

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
