# Peer review of "Experimental Study on Purification Effect of Biochemical Pool Model for Treatment of Pavement Runoff by Aquatic Plants"

_sustainability, doi:10.3390/su12062428_

Round 1

Reviewer 1 Report

In this manuscript, the removal of the pollutants in synthetic runoff by a biochemical pool was investigated. Results showed that the type of plants (Iris pseudacorus, Myriophyllum Verticillatum, and combinations of them) and planting density affected significantly to the removal of COD, petroleum, SS, and metals. Overall, the topic is not very new but interesting, some misused terms and grammatical errors are found, and the results are provided but the discussion was poor. his manuscript should be significantly improved before publication. Specific comments follow:

Line(s) 15-17. I do not understand “. To improve the efficiency … the aquatic plants on the purification.”

Line(s) 17-18 and afterwards, “Iris pseudacorus”, “Myriophyllum verticillatum”, and so on. The scientific names of species must be italicized.

Line(s) 30, “will deposited”. I think it is a grammatical error.

Line(s) 32, “Many researches has been carried”. I think it is a grammatical error.

Line(s) 59, “GI”. Abbreviations should be spelled–out in their first appearance. Is it Green Infrastructure?

Line(s) 100. Please delete “suspended solids”.

Line(s) 108-125. Please shorten this part. The explanation is too lengthy.

Line(s) 147-152. Do you mean that the sediment, generally called as “road deposited sediment (RDS)”, was collected in a range of 0 to 50 cm from the curb or at 50 cm from the curb? According to the reference [33] and another article (Environ. Sci. Pollut. Res. 26, 2019, 1192–1207), the former is right.

Line(s) 147-167. This can be shortened much with adding references because this is a typical procedure of collecting RDS.

Line(s) 140-142. Why did you use “road surface sediments (must be called as RDS)” to prepare feed water? The results showed that the concentration of COD, metals, and petroleum were substantially high. If they were originated from RDS, please provide the analytical results of the RDS used, or at least, add some discussion with references about the pollutants associated with the RDS.

Line(s) 227. Please add how to calculate the removal efficiency given in figures 4-6. Were they the average removal of the samples (#1, #2, and #3), or based on event mean concentration (EMC)?

Line(s) 228-236. Please add reference(s).

Line(s) 239. What are difference between “self-made original runoff sewage” and “experimental runoff water”?

Figures 4-5. It is confusing to interpret the vertical bars (change in purification concentration). Do they mean that the top of the bars is initial concentration and the bottom of the bars is effluent concentration? Then, I do not think that the term “change in purification concentration” is proper. Please revise the figures or legends.

Figures 4-6. Please replace the figures with those of higher resolution.

Line(s) 295-303. What if the removal was converted to removed pollutant mass per (dry) mass of the plants? It may provide useful information for future applications.

Line(s) 316-317. Please add more explanation about the “self-purification effect”. It may largely be attributed to sedimentation.

Line(s) 347. I think the study discussed in the section 3.2 is rather a kinetic study than a retention time effect.

Line(s) 368-370, “During the experiment, after … water to form a weakly acidic substance,”. Please add reference(s) to support this explanation.

Reviewer 2 Report

Reviewer #1: Some comments, suggestions, and recommendations:

Major Comments:

1- This manuscript, as submitted, needs major revisions with regards to writing in English, there are numerous errors in grammar

2- Figures quality needs to be improved, most of the photo labels are not clear  

3- Lack of references in several sentences

Section 1: Abstract and Introduction

Generally, the introduction part needs a lot of improvement, the authors use literature to list serval studies that were conducted on pollutant removal from urban stormwater runoff that is less relevant to the scope of the publication. Are there other papers addressing the removal of the SS, heavy metals and COD using different plant densities? What did they find? There are several studies that were recently published similar to the author’s research and the authors can include in the introduction.

Line 30-31: “For example, the highway surface runoff, which was after the formation of rainfall, will deposited on the road surface as the carrier of pollutants and hurt the affected water environment” the sentence is not clear.

Line 32-33: “Thon et al. [3] proposed a stochastic approach in determining the appropriate wetland size…..” it is better to define the biochemical pool and how it is different from the wetland, bioretention, bioswale, and other stormwater management practices.

Line 32-43: The authors only mentioned the previous studies only without highlighting the results obtained from each one “Thon et al., Stotz G., Drapper et al., Gan et al. and Zhao”

Line 44-66: There is a lack of unity in this paragraph, there are many studies mentioned some are lab or field experiments while others are model work; without showing a clear connection between them or the major objective of each study.

Line 50-51: “The Source Loading and Management Model for Windows (WinSLAMM) was used by Young et al. [12] to determine the efficiency of the model in the swale runoff reduction performance.” I don`t think it necessary the model results in the introduction, it is better to focus on the previous field studies.

Line 95-97: “In this research, the biochemical pool of aquatic plants in the combined control system of road runoff was taken as an example, the indoor scaled model pool was made and different aquatic plants were planted in the model pool with different densities” the objective of the study is not clear, please review.

Section 2: Study Plan

This section has a lot of unnecessary details and it was difficult to follow and on several occasions, no reference was cited.  I suggest the experimental details be summarized and listed in a table so it can be easy to understand the experiment different parameters. The plant scientific name should be in italic font. I encourage the authors to work on it.

Line 95-97: “After diffusion through the root itself, the plant's roots provide an aerobic microenvironment for aerobic microorganisms to survive” please check this sentence.

Line 114: I think the presence of equations [1] and [2] are not necessary both have the same meaning.

Line 122-125: “Aquatic plants can absorb a large number of inorganic phosphorus and inorganic nitrogen through photosynthesis…..” please provide a reference for this sentence.

Line 134-138: “Myriophyllum verticillatum belongs to the perennial dicotyledonous submerged plant of haloragaceae, distributed in north and south China …..” please provide a reference for this sentence.

Line 198-199: “The way to purify the water during rainfall is mainly through the physical adsorption of plants and preventing the diffusion of sewage pollutants in the pond.” This statement is not accurate and needs to be checked and justified.

Line 263: Equation [3] needs to be rewritten, it is not clear.

Section 3: Results and Discussion

The results part was considered more organized in comparison to the previous two sections. But I had difficulty trying to understand the figure labels and results sections need more details such as comparing the author's results to previous research studies. Also, the authors stated the results without providing adequate explanations and analyzing the data.

Line 271: “The runoff wastewater was prepared by adding 300 g of collected road sediment into 160 L of tap water.” This statement belongs to the methods section.

Figures 4 and 5: It would be better to include the pollutant name on each figure instead of mentioning it below the figure only in the figure caption.

Round 2

Reviewer 1 Report

I think the authors elaborated much to improve the original manuscript. But some errors should be corrected.

Line(s) 34. Please delete “G.”

Line(s) 132. “founded” must be “found”

Line(s) 238. Please revise equation (1).

Line(s) 239. Please use the lower case for “Where”. Please do not indent this sentence. An equation and the explanation on the symbols are treated as one sentence.

Figures 4-6. Please replace the figures with those of higher resolution. One method is to copy the plots and paste them as meta files if Microsoft office was used. Please delete the titles of (a)-(f). They were assigned in the figure captions.

Reviewer 2 Report

The revised version was greatly improved but there are still some minor comments that can be addressed.

Section 1: Abstract and Introduction:

I still recommend the author`s to define the biochemical pool, since this term is less commonly used in stormwater treatment. They have already mentioned Green Infrastructure (GI) systems such as wetland and bioretention but didn`t define what is a biochemical pool is? It seems to have the same removal concepts and similar design. What is the difference between a biochemical pool and a wetland? I think this will help the readers to understand more their study.

Line 44-45: “When it comes to pollution, a lack of information on the load and characteristics of pollutants has led to insufficient reduction measures.” Please provide a reference to this sentence.

Line 85-86: “. Blanco [13] indicated totora’s capacity to withstand high concentrations of a cocktail of multiple pollutants and heavy metals.” No need for the word “totora’s” to be italicized and delete the word “cocktail” you can replace it with the word “mixture”.

Line 85-86: “Tang et al. [27] studied nitrogen and phosphorus removal of 7 different aquatic plants including acorus gramineus in small constructed wetlands.” Should be “Acorus” with a capital “A”.

Section 2: Study Plan

I still suggest the experimental details be summarized and listed in a table so it can be easy to understand the experiment different parameters including the size and species.

Line 235: “PH” in “Table 1” should be “pH” please change it throughout the manuscript.

Line 238: Equations [1] is duplicated, please delete one of them.

Section 3: Results and Discussion

Still, the resolution of the figure is low and needs to be improved.

Line 265: “(d) petroleum purification effect.” “P” in petroleum should be capitalized in the figure also.
